# The Comprehensive Machine Learning Analytics for Heart Failure

**DOI:** 10.3390/ijerph18094943

**Published:** 2021-05-06

**Authors:** Chao-Yu Guo, Min-Yang Wu, Hao-Min Cheng

**Affiliations:** 1Institute of Public Health, School of Medicine, National Yang-Ming University, Taipei 112, Taiwan; ben1423@hotmail.com.tw; 2Institute of Public Health, School of Medicine, National Yang Ming Chiao Tung University, Hsinchu 300, Taiwan; 3Center for Evidence-Based Medicine, Veteran General Hospital, Taipei 112, Taiwan; 4Department of Medicine, School of Medicine, National Yang Ming Chiao Tung University, Hsinchu 300, Taiwan

**Keywords:** heart failure, machine learning, prediction model, LASSO logistic regression, support vector machine, random forest, XGBoost

## Abstract

*Background*: Early detection of heart failure is the basis for better medical treatment and prognosis. Over the last decades, both prevalence and incidence rates of heart failure have increased worldwide, resulting in a significant global public health issue. However, an early diagnosis is not an easy task because symptoms of heart failure are usually non-specific. Therefore, this study aims to develop a risk prediction model for incident heart failure through a machine learning-based predictive model. Although African Americans have a higher risk of incident heart failure among all populations, few studies have developed a heart failure risk prediction model for African Americans. *Methods*: This research implemented the Least Absolute Shrinkage and Selection Operator (LASSO) logistic regression, support vector machine, random forest, and Extreme Gradient Boosting (XGBoost) to establish the Jackson Heart Study’s predictive model. In the analysis of real data, missing data are problematic when building a predictive model. Here, we evaluate predictors’ inclusion with various missing rates and different missing imputation strategies to discover the optimal analytics. *Results*: According to hundreds of models that we examined, the best predictive model was the XGBoost that included variables with a missing rate of less than 30 percent, and we imputed missing values by non-parametric random forest imputation. The optimal XGBoost machine demonstrated an Area Under Curve (AUC) of 0.8409 to predict heart failure for the Jackson Heart Study. *Conclusion*: This research identifies variations of diabetes medication as the most crucial risk factor for heart failure compared to the complete cases approach that failed to discover this phenomenon.

## 1. Introduction

The incidence and prevalence of heart failure have been increasing in recent years, and this phenomenon has become a serious global public health issue [1]. Research has estimated that more than 5.8 million people are suffering from heart failure in the United States, and this is increasing at a rate of at least 550,000 per year, while in the world, more than 23 million people are suffering from heart failure [2]. Other studies also pointed out that African Americans have a higher incidence of heart failure than other races [3,4], and even before the age of 50, the risk of heart failure of African Americans is 20 times that of whites [4].

Machine learning is equipped with algorithms that improve performance with experience [5]. A machine learns if its performance at a task improves with experience [6]. In constructing a model using machine learning techniques, the first step divides the entire data into two parts: the training data and the testing data. The training data refers to the dataset used to train the machine learning hyperparameters. The optimized model parameters are estimated based on the training datasets. The testing data is independent of the training data. The testing data will not participate in the training process but will only be used to evaluate the trained model. To avoid overfitting [7], K-Fold Cross Validation (CV) is a common strategy. The CV errors based on accuracy, Mean Square Error (MSE), and F1 Score are adopted [8].

Although it has been indicated that African Americans have a higher risk of incident heart failure among all populations [3], few studies have developed a risk prediction model for heart failure in African Americans. Therefore, this study utilized the Jackson Heart Study (JHS), the most prominent African American research database in the United States, and machine learning methods to construct comprehensive predictive models for heart failure. Our large-scale analysis structure aimed to discover missing and yet important information in such a big-data approach.

Random forest can provide feature importance through decision trees [9]. Breiman discussed the complete algorithm in 2001 [10], a type of ensemble method that collects multiple weak classifiers that produce a robust classifier [11]. The Classification and Regression Tree (CART) is a decision tree for predictive classification and continuous value [12] that adopts the binary division rule. Each time a division generates two branches, and the Gini classification method determines which branch is the best. Extreme Gradient Boosting (XGBoost) is a further extension from GBDT (Gradient Boosted Decision Tree) [13]. 

After missing data are correctly imputed, four methods, including (1) the Least Absolute Shrinkage and Selection Operator (LASSO) logistic regression, (2) Support Vector Machine (SVM) [6,7], (3) random forest, and (4) Extreme Gradient Boosting (XGBoost), would be constructed for a predictive model for heart failure based on the African American population. The inclusion of the four strategies considers the essential aspects of the analysis concepts. LASSO logistic regression is the conventional statistical approach with a penalty term $\lambda \overset{p}{\underset{j=1}{\sum }}\left|\beta_{j}\right|\;$ that avoids overfitting in regression models. The SVM is a machine learning tool dealing with classification problems. The SVM classifies subjects according to the separating hyperplane, which is defined as $\omega^{T}x_{i}+b=0$, where $\omega^{T}x_{i}+b\geq 1,\forall y_{i}=1$ and $\omega^{T}x_{i}+b\leq -1,\forall y_{i}=-1$. The last two methods are tree-based machine learning. However, the random forest and XGBoost are based on different mathematical optimizations. Random forest is an ensemble learning method that constructs a multitude of decision trees at training time. However, the XGBoost depends on gradient boosting trees, and the objection function is defined as $Obj^{\left(t\right)}=\overset{T}{\underset{j=1}{\sum }}\left[\left(G_{j}\right)w_{j}+\frac{1}{2}\left(H_{j}+\lambda \right)w_{j}^{2}\right]+\gamma T$, where $G_{j}=\underset{i\in I_{j}}{\sum }g_{i}$ is the sum of the first derivative of the loss function in the jth leaf and $H_{j}=\underset{i\in I_{j}}{\sum }h_{i}$ represents similar calculations for the second derivative.

As a result, we examined 112 scenarios, and Figure 1 displays the analysis plan. To date, this research had the most extensive models, with seven missing patterns for quality control, four missing imputation strategies, and four power machine learning models. We aimed to discover novel indications for heart failure from such a big scale of analysis models.

## 2. Materials and Methods

Following the previous work [14], the JHS initially included 3883 people. After excluding those who had experienced heart failures and those who were unsure whether they had heart failures or not at the baseline, 3327 people remained in the study. At the end of the study, 246 incident heart failure cases were identified. Table 1 shows the descriptive statistics. Subjects with heart failures accounted for 7.4% of the overall study sample.

Regarding the missing data, variables in the database were classified according to the missing rates, which were <1%, <3%, <5%, <10%, <20%, <30%, and <40%. The missing rate was specific to the variable but not the proportion of missing data in the whole dataset. The column (<40%) means that this dataset contained all variables with missing rates less than 40%. Therefore, more variables were included in the analyses compared to other missing rates. Four imputation techniques dealt with the missing values, including (1) complete cases, (2) simple imputation, (3) the K Nearest-Neighbor (KNN) interpolation method [15], and (4) random forest imputation. More details of the imputation strategy and the statistical properties are available in a former article [16]. Besides, the random forest interpolation method is a machine learning-based imputation of the improved “MissForest” interpolation method [17].

After missing values were imputed, the next step was to generate dummy variables for the unordered category variables. This research adopted the Python language with “one-hot encoding” for categorical variables. We normalized continuous variables according to Min-Max normalization. The formula is $x^{\prime }=\frac{x-\min\left(x\right)}{\max\left(x\right)-\min\left(x\right)}$. The Min-Max normalization limited the range of each variable in the dataset to [0, 1]. In this way, each variable had the same scale, which could improve the predictive performance by avoiding an extremely skewed distribution. Finally, we divided the dataset into the training set (70%) and the testing set (30%). Figure 2 demonstrates the flow chart of the entire data processing.

Note that subjects with heart failures in this study population accounted for 7.4% of the overall sample. For machine learning models, this is an imbalance problem that introduces biases when building the model. We controlled the “class weight” in the LASSO logistic regression, SVM, and random forest to avoid potential biases. The “class weight” parameter in this paper is “balanced,” and the formula is: $w_{j}=\frac{n}{kn_{j}}$, where “$w_{j}$” represents the weight of the jth category of the sample, “n” represents the sample size, “k” represents the number of categories, and “*n_j_*” represents the sample size of the jth category. In the XGBoost, increasing the weight by adjusting the parameter “scale_pos_weight” could increase the predictive ability. The “scale_pos_weight” parameter finds the most appropriate value through K-fold cross-validation, and its range is between 1 and 10.

The logistic regression model is estimated by maximum likelihood [18], but we implemented the Least Absolute Shrinkage and Selection Operator (LASSO)-based model in this research [19]. The SVM [20] with various kernel functions were considered, including the linear kernel, polynomial kernel, and radial basis function kernel. Ten-fold cross-validation found the most suitable kernel function to establish the optimal SVM to predict heart failure.

This research’s XGBoost model is based on 10-fold cross-validation to find the best parameters to establish a heart failure prediction model. The parameters and ranges are as follows: “scale_pos_weight”: range 1~10. “n_estimators”: initial training 90~150, final training 500~3500, “max_depth”: range 2~10. “min_child_weight”: range 1~10. “Gamma”: range 0~5. “Subsample”: range 0.3~1. “colsample_bytree”: range 0.3~1. “reg_lambda”: range $10^{-3}$~$10^{2}$. “reg_alpha”: range $10^{-3}$~$10^{2}$. “learning_rate”: range 0.01~0.1. As there are too many XGBoost parameters, it is time-consuming to train all the parameters simultaneously. Therefore, our procedure adopted training one set of two sets of parameters and gradually trained the XGBoost model (Figure 3).

## 3. Results

If the analyses excluded missing data, Figure 4 shows the corresponding results, where the Y-axis is Area Under Curve (AUC), and the X-axis is the proportion of the missing value of the data. The predictive ability of XGBoost was better than the other three in most scenarios, and the best AUC was 0.808 when the missing rates of included variables were less than 3%. The best AUC of random forest was 0.8003 when the missing rates were less than 3%. The best AUC of SVM was 0.7892 when the missing rates were less than 3%. The best AUC of LASSO logistic regression was 0.7676 when the missing rates were less than 3%.

In summary, the best overall AUC performance occurs when the missing rates of included variables are less than 3%. When we included more variables in the analysis, we also excluded more subjects from the analysis due to missing values. As a result, the AUC decreased.

If the simple interpolation method is adopted, Figure 5 reveals the corresponding results. Similarly, XGBoost was the best performer, and the highest AUC was 0.8239 when the missing rates were less than 30%. The predictive abilities of random forest and LASSO logistic regression are very similar. The best AUC of random forest was 0.813, and LASSO logistic regression demonstrated a value of 0.814. Lastly, the AUC of SVM was 0.79 on average.

In summary, the best overall AUC performance was when the proportion of missing values of the included variables was less than 30%. If we further included the variables whose missing value ratio was less than 40%, the AUC decreased. Therefore, if a follow-up researcher wants to use a simple interpolation method to establish a prediction model, the suggestion from the results of this paper is to include variables whose proportion of missing values is less than 30%. The prediction performance will be relatively good, of which the best performance was the XGBoost prediction model, which had an AUC of 0.8239.

If the KNN interpolation method is adopted, Figure 6 shows the corresponding results. Among all scenarios, XGBoost remained the best performer, and the highest AUC was 0.8369 if the missing rates were less than 30%. Random forest and LASSO logistic revealed similar performance. The best AUC of random forest was 08136 when the missing rates were less than 30%, and LASSO logistic regression was 0.8157 when the missing rates were less than 20%. Finally, the AUC of SVM was 0.79 if the missing rates were less than 30%. In summary, the best AUC occurs when the missing rates are less than 30%. 

If the random forest interpolation method is adopted, Figure 7 shows the corresponding results. The XGBoost outperformed the other three methods in all analyses, and the best AUC was 0.8409 when the missing rates were less than 30%. Random forest and LASSO logic regression have similar predictive abilities. The best AUC of random forest was 0.8191 when the missing rates were less than 30%, while LASSO logic regression was 0.8166 when the missing rates were less than 20%. Lastly, the AUC of SVM was still about 0.79. In summary, the best AUC occurs when the missing rates are less than 30%.

According to these results, our analysis plan is an excellent example for future studies in various health outcomes. We conclude that even when variables have a missing rate as high as 30%, as long as we adopt the random forest imputation before the analysis the XGBoost provides the best predictive ability.

## 4. Conclusions

In this research, the largest African American database in the United States, the JHS, was comprehensively examined by four different machine learning techniques to establish a heart failure predictive model. We also considered the impact of different missing rates and missing data imputation strategies on the predictive ability to discover the best analytic model. Besides, this work provides a series of guidelines for researchers interested in the JHS for future studies.

Based on these results, we conclude that (1) the XGBoost has the best performance in most situations to predict heart failure, especially when we implemented the random forest interpolation with the missing rates at the category of less than 30%. (2) The predictive performance using imputed data would be better than the models without imputations. Even the simplest imputation outperforms the complete cases scenario. (3) When building a predictive model, including too many variables (missing rates less than 40%) results in a worse prediction. The results revealed that in the analysis of complete cases, including variables with a missing rate of 3%, the predictive model performs better. Regarding the imputed data, it can tolerate up to 30% of the missing rates.

Finally, our work uses the feature importance function in random forest and XGBoost to discover which variables significantly contribute to predicting heart failures. Feature importance scores can provide insight and guidance in building the predictive model. We could use the importance scores to delete variables with the lowest scores and keep the variables with the highest scores.

Since the XGBoost performs better than random forest, we did not show the results of random forest. Table 2, Table 3, Table 4 and Table 5 display details of the XGBoost. If we excluded missing data from the analysis, risk factors could not be identified consistently (Table 2). In contrast, if the analyses incorporated the imputed data, diabetes medication (DMmeds) was the most crucial feature suggested by the XGBoost results regardless of the imputation strategy. In Table 2, for data missing <5%, the feature importance of diabetes medication is 0.0492 compared to the age variable with 0.0290. Thus, diabetes medication is almost two times more important when building the predictive model (0.0492/0.029). DMmeds is a dichotomous variable. In this research, 15.25% of the population had a value of 1 with at least one diabetes medication. In total, 84.75% were without diabetes medication. Therefore, variations of diabetes medication were the most influential risk factor for heart failure.

Furthermore, most of the findings revealed that heart failure was related to age, sex, diabetes, kidney function, and past heart disease history. Previous studies also pointed out that these variables are risk factors for heart failure [21,22,23], suggesting that our prediction models established by machine learning could discover crucial risk factors. In particular, a former study indicated significant predictors, including age, sex, body mass index, diabetes mellitus, systolic blood pressure, creatinine, serum albumin or total protein, LV hypertrophy, and coronary artery disease [24]. Compared to the simple AHEAD scoring system [25], the other AHEAD-U scoring system proposed by Chen et al. [26], and the most recent HANBAH score by Guo et al. [27], the most crucial feature importance variables determined by the XGBoost are similar to these variables used in the scoring system. However, variations of diabetes medications have never been reported previously. Therefore, the XGBoost discovered a novel influential risk factor and other similar factors that match previous studies.

In summary, the results suggest using variables with missing rates of less than 30% and adopting the Random Forest imputations when building a heart failure predictive model using the JHS database. Finally, we recommend that the XGBoost build the risk model, and the highest predictive accuracy is 0.8409. Most importantly, this research identifies variations of diabetes medication as the most crucial risk factor for heart failure, regardless of the missing data imputation strategy, since the variable of diabetes medications has the highest feater importance. In contrast, the commonly adopted approach that only uses complete cases failed to provide such novel findings.

The variable “diabetes medication” indicates that patients had diabetes, which is different from the conventional understanding of diabetic treatment [28]. The superficial understanding suggests that patients receiving diabetic treatment had a worse prognosis than those without treatment. It can be misunderstood that diabetes medication is the most crucial risk factor for heart failure. In this research, we interpret the variable “diabetes medication” as variations of diabetes medication since this variable has multiple levels, and we treated it as a dummy variable in the analysis.

In previous research [29,30], the authors used five methods to predict heart failure, including logistic regression, decision tree, random forest, simple Bayesian classifier, and support vector machine (SVM). The source of the data is from the Cleveland database. The number of samples included was 303, and the number of variables included was 14. For the data’s missing values, the study used the K-Nearest Neighbor (KNN) interpolation method. In the final predictive results, the study adopted accuracy to conclude that the five methods have good prediction accuracy (87.36%, 93.19%, 89.14%, 87.27%, 92.30%), and the decision tree performed best.

Later, the random survival forest [31] built a predictive model. The database was Action to Control Cardiovascular Risk in Diabetes (ACCORD). There were 8756 samples included. After excluding variables with a missing value greater than 10%, the number of variables included was 109. Regarding the missing values, this research imputed missing data using random forest interpolation. The C-index evaluated the final predictive results, and the random survival forest yielded a value of 0.74.

Another study implemented three methods, including logistic regression, SVM, and Ada-Boost, to predict heart failure [32]. They came from the Geisinger Clinic Electronic Health Record system (Electronic Health Record). The sample size was 4489, and the number of variables was 179. For the missing values, this study used the missing indicator method. For model optimization and selection of parameters, this study adopted a 10-fold CV. The study used the Area Under the Curve (AUC) to compare the final predictive results. The Receiver Operating Characteristic (ROC) curve is an excellent diagnostic tool [33]. The Youden Index is also a popular choice [34] defined as  argmax(true positive rate +true negative rate−1, 1). The higher Area Under the Curve (AUC) [35] also indicates a better prediction [36]. Previous research suggested the interpretations of different cutoffs of AUC higher than 0.5 [37]. The logistic regression results were similar to the Ada-Boost with an AUC of 0.77, and the SVM was relatively unsatisfactory with an AUC of 0.62.

In future development, one could further discuss the following aspects. The problem of imbalanced data explains the weight of a small number of samples to be adjusted. There are many methods to deal with this issue, such as undersampling, which reduces multiple types of samples, or oversampling (SMOTE [38]) to generate more small samples, such that a ratio of normal and diseased samples of 1:1.

Other methods can build predictive models, such as the artificial neural network, naive Bayes, or other ensemble algorithms. These areas are essential topics in the future.

## Figures and Tables

**Figure 1 ijerph-18-04943-f001:**
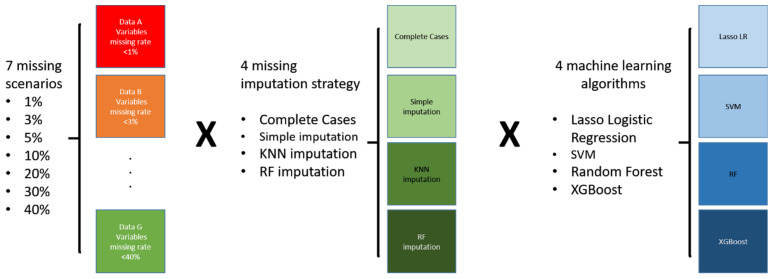
Scenarios considered.

**Figure 2 ijerph-18-04943-f002:**
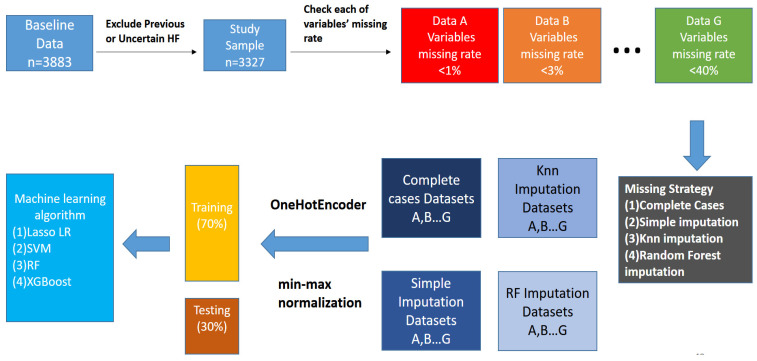
Study flow chart.

**Figure 3 ijerph-18-04943-f003:**
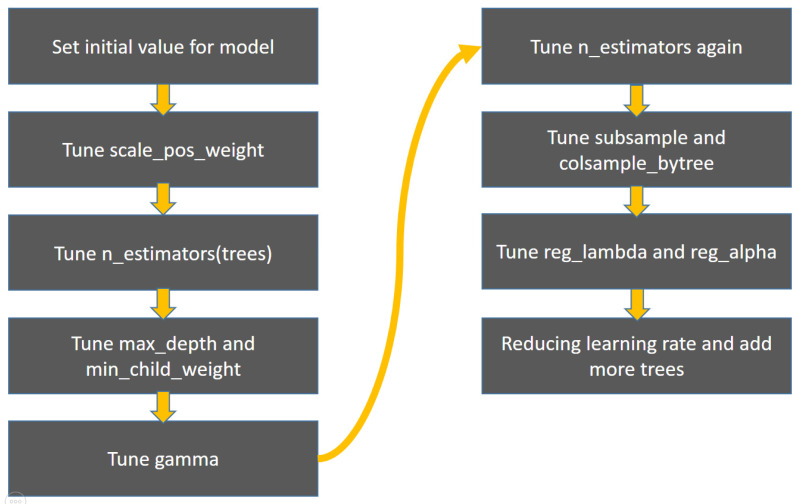
The training process of XGBoost.

**Figure 4 ijerph-18-04943-f004:**
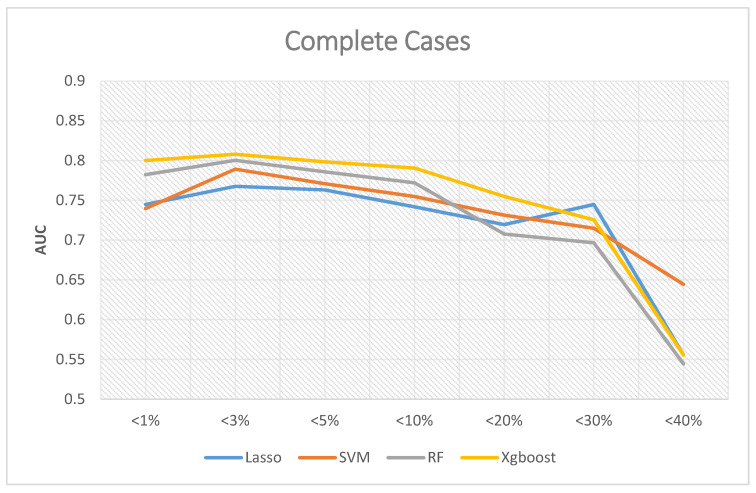
AUC for complete cases.

**Figure 5 ijerph-18-04943-f005:**
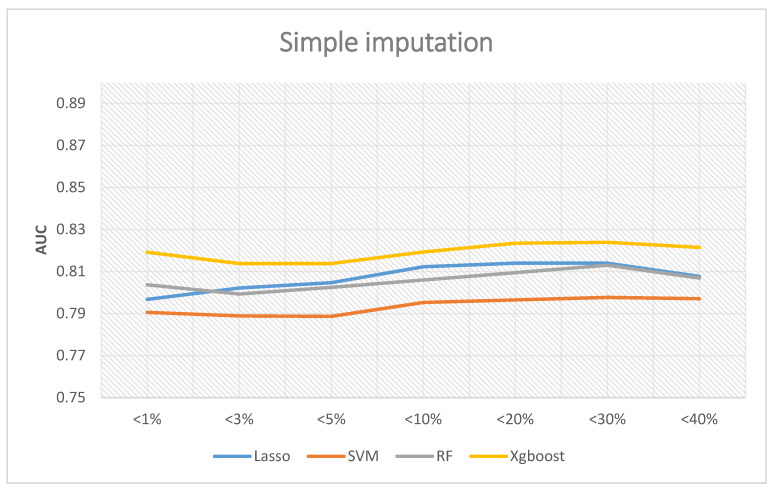
AUC with simple imputation.

**Figure 6 ijerph-18-04943-f006:**
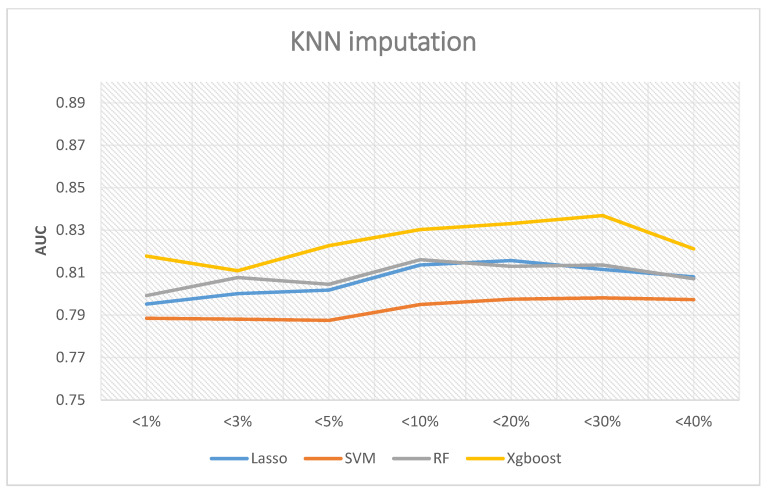
AUC with KNN imputation.

**Figure 7 ijerph-18-04943-f007:**
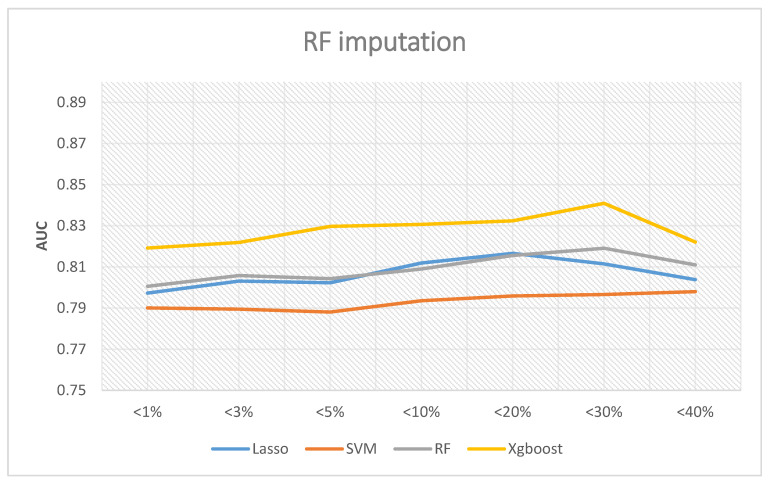
AUC with RF imputation.

**Table 1 ijerph-18-04943-t001:** Descriptive statistics of the study population.

Baseline Characteristic	Total Population *n* = 3327	Non-HF *n* = 3081 (92.6%)	HF *n* = 246 (7.4%)
Age	54.96 (12.59)	54.24 (12.37)	63.91 (11.98)
BMI	31.82 (7.2)	31.72 (7.18)	33.02 (7.31)
Waist	100.83 (16.03)	100.4 (15.93)	106.9 (16.14)
High School Graduate	2761 (83.26%)	2607 (84.92%)	154 (62.60%)
Gender			
Male	1228 (36.91%)	1132 (36.74%)	96 (39.02%)
Female	2099 (63.09%)	1949 (63.26%)	150 (60.98%)
Current Smoker	406 (12.31%)	374 (12.25%)	32 (13.11%)
Hypertension (HTN)	1845 (55.47%)	1644 (53.38%)	201 (81.71%)
Diabetes Mellitus (DM)	710 (21.5%)	593 (19.39%)	117 (47.95%)

**Table 2 ijerph-18-04943-t002:** The selected parameters of the final HF prediction model for XGBoost using complete cases (feature importance is in the parenthesis).

<1%	<3%	<5%	<10%	<20%	<30%	<40%
age (0.0923)	age (0.0213)	DMmeds (0.0492)	DMmeds (0.0222)	dmMeds (0.0178)	CVDHx (0.0267)	frs_chdtenyrrisk (0.0507)
DMmeds (0.0647)	RepolarAntLat (0.0189)	age (0.0290)	Diabetes (0.0192)	MIHx (0.0174)	ascvd_tenyrrisk (0.0260)	ALDOSTERONE (0.0367)
Diabetes (0.0431)	DMmeds (0.0187)	BP3cat (0.0270)	CVDHx (0.0189)	EF (0.0170)	rrs_tenyrrisk (0.0220)	eGFRmdrd (0.0352)
eGFRckdepi (0.0315)	Diabetes (0.0180)	HTN (0.0267)	bpjnc7_3 (0.0187)	HbA1cIFCC (0.0150)	numbnessEver (0.0203)	occupation (0.0331)
MIecg (0.0305)	CVDHx (0.0174)	sbp (0.0237)	CHDHx (0.0176)	HbA1c (0.0148)	nutrition3cat (0.0194)	abi (0.0317)
RepolarAntLat (0.0297)	eGFRmdrd (0.0173)	eGFRckdepi (0.0233)	eGFRckdepi (0.0174)	strokeHx (0.0141)	FEV1PP (0.0193)	sbp (0.0297)
antiArythMeds (0.028)	eGFRckdepi (0.0173)	CVDHx (0.0188)	FPG (0.0170)	Diabetes (0.0141)	totchol (0.0186)	calBlkMeds (0.0292)
RepolarAnt (0.0272)	statinMeds (0.0168)	QTcFrid (0.0187)	age (0.0170)	visionLossEver (0.0140)	asthma (0.0180)	FVC (0.0289)
statinMeds (0.0268)	edu3cat (0.0163)	BPmeds (0.0184)	sbp (0.0164)	statinMeds (0.0129)	BPmeds (0.0177)	eGFRckdepi (0.0253)
CVDHx (0.0262)	CardiacProcHx (0.0162)	waist (0.01831)	waist (0.0160)	frs_chdtenyrrisk (0.0129)	HTN (0.0177)	SCrCC (0.0242)

Note: The above abbreviations are available in Table A1.

**Table 3 ijerph-18-04943-t003:** The selected parameters of the final HF prediction model for XGBoost using simple imputations.

<1%	<3%	<5%	<10%	<20%	<30%	<40%
Diabetes (0.0455)	DMmeds (0.0346)	DMmeds (0.0588)	DMmeds (0.0233)	DMmeds (0.0630)	DMmeds (0.0210)	DMmeds (0.0337)
DMmeds (0.0439)	age (0.0273)	Diabetes (0.0327)	age (0.0221)	age (0.0383)	age (0.0196)	DialysisEver (0.0252)
age (0.0408)	CVDHx (0.0265)	DialysisEver (0.0243)	eGFRckdepi (0.0185)	Diabetes (0.0297)	Diabetes (0.0158)	CVDHx (0.0207)
HTN (0.0336)	eGFRckdepi (0.0263)	MIAntLat (0.0191)	Diabetes (0.0183)	DialysisEver (0.0190)	BPmeds (0.0150)	age (0.0151)
CVDHx (0.0277)	HTN (0.0260)	age (0.0189)	FEV1 (0.0163)	ConductionDefect (0.0183)	CVDHx (0.0147)	Afib (0.0142)
HSgrad (0.0273)	HSgrad (0.0246)	HSgrad (0.0157)	CVDHx (0.0158)	sex (0.0180)	age (0.0145)	MIHx (0.0136)
BPmeds (0.0272)	Diabetes (0.0237)	Afib (0.0148)	FVC (0.0156)	occupation (0.0159)	ConductionDefect (0.0141)	eGFRckdepi (0.0129)
eGFRckdepi (0.0238)	eGFRmdrd (0.0225)	edu3cat (0.0138)	CHDHx (0.0152)	MIant (0.0143)	FEV1 (0.0141)	SystLVdia (0.0128)
RepolarAntLat (0.0229)	MIHx (0.0212)	CVDHx (0.0135)	eGFRmdrd (0.0151)	CVDHx (0.0129)	eGFRckdepi (0.0139)	EF (0.0117)
ecgHR (0.0206)	sbp (0.0191)	EF (0.0130)	HbA1cIFCC (0.0148)	idealHealthSMK (0.0125)	CHDHx (0.0126)	ConductionDefect (0.0115)

**Table 4 ijerph-18-04943-t004:** The selected parameters of the final HF prediction model for XGBoost using KNN imputations.

<1%	<3%	<5%	<10%	<20%	<30%	<40%
Diabetes (0.0435)	age (0.0320)	DMmeds (0.0848)	DMmeds (0.0261)	DMmeds (0.0443)	DMmeds (0.0145)	DMmeds (0.0318)
DMmeds (0.0409)	DMmeds (0.0266)	age (0.0302)	age (0.0177)	age (0.0420)	age (0.0142)	DialysisEver (0.0294)
age (0.0386)	MIHx (0.0234)	MIant (0.0245)	CVDHx (0.0175)	CVDHx (0.0321)	eGFRmdrd (0.0138)	age (0.0234)
HTN (0.0299)	Diabetes (0.0213)	CVDHx (0.0241)	eGFRckdepi (0.0173)	EF (0.0188)	Diabetes (0.0130)	Diabetes (0.0145)
CVDHx (0.0277)	CVDHx (0.0209)	Diabetes (0.0236)	Diabetes (0.0167)	eGFRckdepi (0.0182)	SCrCC (0.0122)	Afib (0.0143)
BPmeds (0.0274)	eGFRckdepi (0.0205)	HSgrad (0.0206)	eGFRmdrd (0.0153)	FEV1 (0.0179)	eGFRckdepi (0.0118)	CVDHx (0.0128)
HSgrad (0.0264)	HSgrad (0.0176)	ConductionDefect (0.0201)	FEV1 (0.0153)	ConductionDefect (0.0174)	statinMeds (0.0115)	eGFRckdepi (0.0126)
eGFRckdepi (0.0222)	HTN (0.0172)	antiArythMedsSelf (0.0152)	CHDHx (0.0152)	MIHx (0.0172)	CVDHx (0.0112)	MIHx (0.0120)
RepolarAntLat (0.0209)	antiArythMeds (0.0171)	CHDHx (0.1431)	edu3cat (0.0142)	MajorScarAnt (0.0172)	everSmoker (0.0111)	calBlkMeds (0.0116)
ecgHR (0.0196)	eGFRmdrd (0.0164)	AntiArythMeds (0.1374)	DialysisEver (0.0139)	eGFRmdrd (0.0170)	rrs_tenyrrisk (0.0108)	FEV1 (0.0113)

**Table 5 ijerph-18-04943-t005:** The selected parameters of the final HF prediction model for XGBoost using MissForest imputations.

<1%	<3%	<5%	<10%	<20%	<30%	<40%
Diabetes (0.0455)	antiArythMeds (0.0339)	dmMeds (0.0340)	DMmeds (0.0263)	DMmeds (0.0443)	DMmeds (0.0163)	DMmeds (0.0290)
DMmeds (0.0363)	DMmeds (0.0332)	age (0.0279)	age (0.0251)	DialysisEver (0.0323)	Diabetes (0.0161)	ascvd_tenyrrisk (0.0255)
age (0.0344)	age (0.0301)	Diabetes (0.0271)	Diabetes (0.0234)	MIAntLat (0.0241)	rrs_tenyrrisk (0.0148)	age (0.0218)
HTN (0.0336)	eGFRckdepi (0.0241)	eGFRckdepi (0.0269)	CVDHx (0.0218)	Diabetes (0.0208)	age (0.0132)	eGFRckdepi (0.0210)
CVDHx (0.0318)	HTN (0.0232)	CVDHx (0.0235)	CHDHx (0.0194)	age (0.0194)	ascvd_tenyrrisk (0.0130)	rrs_tenyrrisk (0.0191)
HSgrad (0.0274)	SCrIDMS (0.0220)	eGFRmdrd (0.0212)	eGFRmdrd (0.0192)	Afib (0.0184)	MIant (0.0127)	frs_cvdtenyrrisk (0.0179)
eGFRckdepi (0.0243)	MIHx (0.0207)	HSgrad (0.0198)	eGFRckdepi (0.0162)	calBlkMeds (0.0154)	eGFRckdepi (0.0125)	MIHx (0.0162)
CHDHx (0.0241)	CVDHx (0.0198)	SCrIDMS (0.0187)	HSgrad (0.0162)	CVDHx (0.0152)	CVDHx (0.0118)	LEPTIN (0.0147)
RepolarAntLat (0.0238)	eGFRmdrd (0.0197)	BPMeds (0.0170)	FEV1 (0.0149)	eGFRckdepi (0.0149)	FEV1 (0.0106)	calBlkMeds (0.0135)
QTcBaz (0.0221)	Diabetes (0.0195)	HbA1c (0.0158)	SCrIDMS (0.0148)	EF (0.0140)	CHDHx (0.0104)	CardiacProcHx (0.0127)

## Data Availability

This manuscript was prepared using the database of Jackson Heart Study with materials obtained from the NHLBI Biologic Specimen and Data Repository Information Coordinating Center and does not necessarily reflect the opinions or views of the Jackson Heart Study or the NHLBI. The views expressed in this manuscript are those of the authors and do not necessarily represent the views of the National Heart, Lung, and Blood Institute; the National Institutes of Health; or the U.S. Department of Health and Human Services.

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
