# Peer review of "The Comprehensive Machine Learning Analytics for Heart Failure"

_ijerph, 2021, doi:10.3390/ijerph18094943_

Round 1
Reviewer 1 Report
The paper needs significant revision before it can be considered for publication.
- Please describe all acronyms before first usage.
- The title while catchy should reflect more on the methodology than the results.
- Throughout the paper non-specific terms like 'may', 'generally', etc. are use. Please be exact in assertions and provide suitable references.
- Since imputations play such a major role in the study, an overview of the four imputation methods with examples is required.
- For Table 1 it is unclear how the data is divided into HF and non HF when patients with HF were were excluded.
- Figures 4-7 require major improvements. Please label axes, add notches and outlines. Tables, plots and data labels is too much information (with a lot of duplication) and must be simplified.
- It is rather puzzling that in figures 5-7 the fraction of missing data makes very little difference to the AUC. Please check for statistical significance. Comment why this might be the case.
- A lot of the material in Results is actually Discussion (e.g. page 7, first paragraph).
- I think Tables 2-5 would benefit from simplification. Use proper feature names and acronyms and perhaps a color bar to denote fraction. Listing the top few features should be sufficient.
- While diabetes medications is consistently at the top of the list, please find a better way to quantify its importance. Importance of features over all four tables should be looked at individually as well as together.
- JHS is not the largest dataset. Please take a look at MIMIC. Would your study generalize to larger data sets?
- The discussion fails to describe the true impact of the paper. Please expand considerably and discuss innovation, knowledge gaps, impact, limitations and future directions.
Author Response
Reviewer 1:
The paper needs significant revision before it can be considered for publication.
Please describe all acronyms before first usage.
The title while catchy should reflect more on the methodology than the results.
Reply No. 1.1:
We sincerely appreciate the kind response and the excellent suggestion for the title. The revision has a new, clearer, and shorter title.
Changes No 1.1:
The new title is "The Comprehensive Machine Learning Analytics for Heart Failure".
Throughout the paper non-specific terms like 'may', 'generally', etc. are use. Please be exact in assertions and provide suitable references.
Reply No. 1.2:
We thank the reviewer for the comment. We had corrected these non-specific terms and the suitable references are cited properly at the end of these statements.
Changes No. 1.2:
New references:
- Swedberg K. Heart failure subtypes: pathophysiology and definitions. Diabetes Res Clin Pract. 2021 Apr 13:108815. doi: 10.1016/j.diabres.2021.108815. Online ahead of print.PMID: 33862057
- Gvozdanović Z, Farčić N, Šimić H, Buljanović V, Gvozdanović L, Katalinić S, Pačarić S, Gvozdanović D, Dujmić Ž, Miškić B, Barać I, Prlić N. The Impact of Education, COVID-19 and Risk Factors on the Quality of Life in Patients with Type 2 Diabetes. Int J Environ Res Public Health. 2021 Feb 27;18(5):2332. doi: 10.3390/ijerph18052332. PMID: 33673454; PMCID: PMC7956830.
Since imputations play such a major role in the study, an overview of the four imputation methods with examples is required.
Reply No. 1.3:
We understand the importance of imputations and the details are available in the two technical referencves. Since this research aims on the discovery of risk factor, we diddn’t put much emphasis on explanations of all methods implemented in this study since there are too many methods involved in the analyses.
Changes No. 1.3:
We updated the following sentences on page 3 so that the details of imputations are directed into the previous researches:
Four imputation techniques would deal with the missing values, including (1) complete cases, (2) simple imputation, (3) the K nearest-neighbor (KNN) interpolation method [15], and (4) random forest imputation. More details of the imputation strategy and the statistical properties are available in the former article [16]. Besides, the random forest interpolation method is a machine learning-based imputation of the improved "MissForest" interpolation method [17]
For Table 1 it is unclear how the data is divided into HF and non HF when patients with HF were were excluded.
Reply No. 1.4:
We apologize for the confusion and added in a more specific description in this paragraph.
Changes No. 1.4:
Following the previous work [14], the JHS initially included 3,883 people. After excluding those who had experienced heart failures and those who were unsure whether they had heart failures or not at the baseline, 3327 people remained in the study. At the end of the study, 246 incident heart failure cases were identified.
Figures 4-7 require major improvements. Please label axes, add notches and outlines. Tables, plots and data labels is too much information (with a lot of duplication) and must be simplified.
Reply No. 1.5:
The comment is well received. The Y axis is labeled with AUC and we illuminate the label for the X axis, because the table on the bottom indicates the exact scenario of the missing rate. In fact, we could present only the table in Figures 4-7. However, we think that the plot shows clear comparisons of the four strategies. The layout of the Figures are the best we could come up with. If the reviewer insist that we change all the figures, we will follow the instructions from the reviewer in the 2nd revision.
It is rather puzzling that in figures 5-7 the fraction of missing data makes very little difference to the AUC. Please check for statistical significance. Comment why this might be the case.
Reply No. 1.6:
We thank the reviewer for this question. On page 3.we stated that “Regarding the missing data, variables in the database are classified according to the missing rates, which are <1%, <3%, <5%, <10%, <20%, <30%, and <40%.”
The missing rate is variable specific but not the proportion of missing data in the whole dataset. The column (<40%) means that this dataset contains all variables with missing rates less than 40%. Therefore, more variables would be included in the analyses comparing to other missing rates.
Changes No. 1.6:
The following sentences are added on page 3 to provide a clearer understanding.
The missing rate is variable specific but not the proportion of missing data in the whole dataset. The column (<40%) means that this dataset contains all variables with missing rates less than 40%. Therefore, more variables would be included in the analyses comparing to other missing rates.
A lot of the material in Results is actually Discussion (e.g. page 7, first paragraph).
I think Tables 2-5 would benefit from simplification. Use proper feature names and acronyms and perhaps a color bar to denote fraction. Listing the top few features should be sufficient.
Reply No. 1.7:
We acknowledged the great comments in this revision, and the two paragraphs have been moved to the concluson section. The original first paragraph has been replaced with 4 paragraphs. Please see the revision for updates.
While diabetes medications is consistently at the top of the list, please find a better way to quantify its importance. Importance of features over all four tables should be looked at individually as well as together.
Reply No. 1.8:
We appreciate the reviewer's comment very much. The numbers behind all the features have quantified the importance. Unlike the statistical approach, a p-value is not available for all features. We believe that we had presented the most important information in the tables.
JHS is not the largest dataset. Please take a look at MIMIC. Would your study generalize to larger data sets?
Reply No. 1.9:
We apologize for the misunderstanding. We stated that it is a largest African-American database, specifically for this ethnicity. We think the information is correct.
The discussion fails to describe the true impact of the paper. Please expand considerably and discuss innovation, knowledge gaps, impact, limitations and future directions.
Reply No. 1.10:
We sincerely appreciate the reviewer's comment. At the end of the second paragraph on page 7, the novelty discovery is emphasized and updated as the following sentences. The conclusions section also pointed out the role of variations of diabetes medications. New references [29,30] is also added to further explain the diabetic treatment.
Changes No 1.10:
At the end of the second paragraph on page 7
However, variations of diabetes medications have never been reported previously. Therefore, the XGBoost discovered a novel influential risk factor and other similar factors that match previous studies.
New references:
- Swedberg K. Heart failure subtypes: pathophysiology and definitions. Diabetes Res Clin Pract. 2021 Apr 13:108815. doi: 10.1016/j.diabres.2021.108815. Online ahead of print.PMID: 33862057
- Gvozdanović Z, Farčić N, Šimić H, Buljanović V, Gvozdanović L, Katalinić S, Pačarić S, Gvozdanović D, Dujmić Ž, Miškić B, Barać I, Prlić N. The Impact of Education, COVID-19 and Risk Factors on the Quality of Life in Patients with Type 2 Diabetes. Int J Environ Res Public Health. 2021 Feb 27;18(5):2332. doi: 10.3390/ijerph18052332. PMID: 33673454; PMCID: PMC7956830.
Reviewer 2 Report
Dear Authors,
Thank you for your work. My concern is the novelty of the work. I don't see any particular contribution to the subject. The accuracy of your findings doesn't seem to be sufficient/significant in the medical domain.
The meaning of "Identify Variations of Diabetes Medication as the Most Crucial Risk Factor for Heart Failure" is not clear to me. What is the meaning of crucial in this context.
It would make much more sense if you could add improvements to current ML methods and report the results.
Self-citations need to be improved.
Author Response
Reviewer 2:
Dear Authors,
Thank you for your work. My concern is the novelty of the work. I don't see any particular contribution to the subject. The accuracy of your findings doesn't seem to be sufficient/significant in the medical domain.
Reply No 2.1:
We sincerely appreciate the reviewer's comment. At the end of the second paragraph on page 7, the novelty discovery is emphasized and updated as the following sentences.
Changes No 2.1:
At the end of the second paragraph on page 7
However, variations of diabetes medications have never been reported previously. Therefore, the XGBoost discovered a novel influential risk factor and other similar factors that match previous studies.
The meaning of "Identify Variations of Diabetes Medication as the Most Crucial Risk Factor for Heart Failure" is not clear to me. What is the meaning of crucial in this context.
Reply No 2.2:
The importance message is well taken. In the statistical language, the most crucial risk factor has the smallest p-value that shows the most significant result. In the first paragraph of conclusion section on page 9, we added a brief highlight of the machine learning results indicating that the variable of diabetes medications has the highest feater importance".
Changes No 2.2:
The paragraph is updated as the following:
Most importantly, this research identifies variations of diabetes medication as the most crucial risk factor for heart failure, regardless of the missing data imputation strategy since diabetes medications has the highest feater importance.
It would make much more sense if you could add improvements to current ML methods and report the results.
Reply No 2.3:
We thank the reviewer for such great advice. We understand that a new machine learning method is highly desired. However, our medical research with a novel discovery also contributes significantly in heart failure research.
Self-citations need to be improved.
Reply No 2.4:
Thanks much for indicating this issue. However, the prediction scores for acute heart failure are crucial in this research topic. Following the AHEAD score, we developed AHEAD-U and HANBAH scores, therefore, the two self-citations are necessary. Please see our response to the first review question 2.1 that emphasizes our novel research findings when building a prediction score for heart failure.
Reviewer 3 Report
In this article, the authors aim to develop a risk prediction model for incident heart failure through a machine learning-based predictive model.
The article is well written with proper readability and length. Minor details on English grammar require review. However, many aspects need to be clarified and improved.
First of all, I suggest the authors revise the title of the paper. The paper's title is too long. Please choose a title that reflects the work conducted in this paper.
The introduction section needs enrichment. The article is missing relevant references to the problem. Also, the authors should mention, briefly, the main contribution of the work.
Please add a separate section entitled "Related Work" to transit smoothly in the following parts of the work. Please evaluate how your study is different from others in the related work section? What you have where others do not? Why you are better or how? What’s the novel here?
The "MATERIALS AND METHODS" section needs to be explained more technical with more details from ML technical points of view. The authors need to elaborate, why the traditional ML classifiers are selected. A background section about ML algorithms and techniques is necessary. Technical reasons must provide in contrast to the state-of-the-art ML approaches. The authors did not explain how these methods used. Please, explain them clearly.
The technical contribution is limited. The author has applied already known techniques and methods.
I suggest the authors, add a table mentioning the parameters tuned for different machine learning algorithms used in this study.
There is no clear presentation of the results and their commentary. Try to make a more coherent, accurate and focused presentation. The presented methodology and the results are not communicated, with the necessary background for the readers included in the paper.
I recommend the authors discuss the merits of the proposed approach.
Please rename the discussion section to "Conclusions". In the conclusion section, the authors must highlight novel propositions, numeric information related to accuracy and other advantages of the proposed solution. Moreover, conclusions can discuss future research directions and extensions of the study.
I recommend the authors include more recent references and from MDPI as well.
To sum up, the article needs an overall revision in terms of the information flow and technique capture.
Author Response
Reviewer 3:
In this article, the authors aim to develop a risk prediction model for incident heart failure through a machine learning-based predictive model.
The article is well written with proper readability and length. Minor details on English grammar require review. However, many aspects need to be clarified and improved.
First of all, I suggest the authors revise the title of the paper. The paper's title is too long. Please choose a title that reflects the work conducted in this paper.
Reply No. 3.1:
We sincerely appreciate the great suggestion and modified the title in the revision.
Changes No 3.1:
The new title is "The Comprehensive Machine Learning Analytics for Heart Failure".
The introduction section needs enrichment. The article is missing relevant references to the problem. Also, the authors should mention, briefly, the main contribution of the work. Please add a separate section entitled "Related Work" to transit smoothly in the following parts of the work. Please evaluate how your study is different from others in the related work section? What you have where others do not? Why you are better or how? What's the novel here?
Reply No. 3.2:
We had added one sentence to emphasize our contribution on page 2 in the introduction and modified the paragraph as well:
Chagnes No. 3.2:
Although it has indicated that African Americans had a higher risk of incident heart failure among all populations [3], few studies have developed the risk prediction model for heart failure in African-Americans. Therefore, this study utilized the Jackson Heart Study (JHS), the most prominent African-American research database in the United States, and machine learning methods to construct comprehensive predictive models for heart failure. Our large-scale analysis structure aims to discover missing and yet important information in such a big-data approach.
The "MATERIALS AND METHODS" section needs to be explained more technical with more details from ML technical points of view. The authors need to elaborate, why the traditional ML classifiers are selected. A background section about ML algorithms and techniques is necessary. Technical reasons must provide in contrast to the state-of-the-art ML approaches. The authors did not explain how these methods used. Please, explain them clearly. The technical contribution is limited. The author has applied already known techniques and methods.
Reply No. 3.3:
We appreciate the comment from the reviewer. We provided references for all the methods used and briefly described the primary conclusion from each reference since this is not a statistical journal, and we want to emphasize more on the results, new findings and conclusions. In this way, a non-statistical audience could easily understand this research. If the readers are interested in more statistical details, these references have all the information.
I suggest the authors, add a table mentioning the parameters tuned for different machine learning algorithms used in this study.
Reply No. 3.4:
The comment is well received. Please see figure 3 for the training process.
There is no clear presentation of the results and their commentary. Try to make a more coherent, accurate and focused presentation. The presented methodology and the results are not communicated, with the necessary background for the readers included in the paper. I recommend the authors discuss the merits of the proposed approach.
Reply No. 3.5:
We truly appreciate the suggestions. Please see our response to the first and second reviewer. We believe that the revision has reflected all these issues.
Please rename the discussion section to "Conclusions". In the conclusion section, the authors must highlight novel propositions, numeric information related to accuracy and other advantages of the proposed solution. Moreover, conclusions can discuss future research directions and extensions of the study.
Reply No. 3.6:
We thank the reviewer for the suggestion.
Changes No. 3.6:
The discussion section has been changed to conclusions.
I recommend the authors include more recent references and from MDPI as well.
To sum up, the article needs an overall revision in terms of the information flow and technique capture.
Reply No. 3.7:
The comment is well received. We had added a new reference from MDPI and the major revision has improved the flow and technical capture.
Changes No. 3.7:
New Reference from MDPI has been added in the conclusions section:
30: Gvozdanović Z, Farčić N, Šimić H, Buljanović V, Gvozdanović L, Katalinić S, Pačarić S, Gvozdanović D, Dujmić Ž, Miškić B, Barać I, Prlić N. The Impact of Education, COVID-19 and Risk Factors on the Quality of Life in Patients with Type 2 Diabetes. Int J Environ Res Public Health. 2021 Feb 27;18(5):2332. doi: 10.3390/ijerph18052332. PMID: 33673454; PMCID: PMC7956830.
Round 2
Reviewer 1 Report
While the authors have addressed most of my comments concerns remain in three areas:
- Figures 4-7 need to be accompanied by statistical analysis and p-values need to be reported.
- Figures 4-7 should be either a plot or a table (not both).
- Quantification of diabetes medication as the most important feature needs to be quantified. A simple metric such as median rank would suffice.
Once addressed, the paper should be publishable.
Author Response
Reviewer 1:
While the authors have addressed most of my comments concerns remain in three areas:Figures 4-7 need to be accompanied by statistical analysis and p-values need to be reported.
Reply No. 1.1
We thank the reviewer for the suggestion. We would do our best to provide the most detailed results. Figures 4-7 show the predictive ability for each method. However, p-value is not available in machine learning strategies.
Figures 4-7 should be either a plot or a table (not both).
Reply No. 1.2:
We sincerely appreciate the great suggestion and modified the four figures.
Changes No. 1.2:
Figure 4:
Figure 5:
Figure 6:
Figure 7:
- Quantification of diabetes medication as the most important feature needs to be quantified. A simple metric such as median rank would suffice.
Reply No. 1.3:
The comment is critical and has been well received.
Changes No. 1.3:
We had added the following sentences in the middle of the first paragraph of the conclusions section on page 10.
In contrast, if the analyses incorporated the imputed data, diabetes medication (DMmeds) was the most crucial feature suggested by the XGBoost results regardless of the imputation strategy. DMmeds is a dichotomous variable. In this research, 15.25% of the population have a value of 1 with at least one diabetes medication. 84.75% are 0 without diabetes medication.
Once addressed, the paper should be publishable.
Reply No. 1.4:
We sincerely appreciate the kind response.

Reviewer 2 Report
Thanks for incorporating the comments into the new version.
Author Response
Reviewer 2:
Thanks for incorporating the comments into the new version.
Reply No. 2.1:
We thank the reviewer for such warm approval.

Reviewer 3 Report
The authors omitted several of my remarks for correction and improvement.
1)The only addition that the authors made to the introduction is "Our large-scale analysis structure aims to discover missing and yet important information in such a big-data approach". I still wish the authors to add a separate section entitled "Related Work" to transit smoothly in the following parts of the work. Please evaluate how your study is different from others in the related work section? What you have where others do not? Why you are better or how? What's the novel here?
2)In a scientific article, it is not enough to cite the references.
The philosophy of the article is relevant to the field of ML.
The authors need to elaborate, why the traditional ML classifiers are selected. A background section about ML algorithms and techniques is necessary. Technical reasons must provide in contrast to the state-of-the-art ML approaches.
3)There is no clear presentation of the results and their commentary
Τhe authors continuously project figures and Tables without highlighting the methodology and the importance of the results. I recommend the authors discuss the merits of the proposed results.
Authors should provide adequate explanations for their methodology in the article.
In a scientific article, each technique should be compared with already known methods, and its superiority should be documented. That is the philosophy of the research. Also, the more details are given about the proposed approach, the better for the reader.
Author Response
Reviewer 3:
The authors omitted several of my remarks for correction and improvement.
1)The only addition that the authors made to the introduction is "Our large-scale analysis structure aims to discover missing and yet important information in such a big-data approach". I still wish the authors to add a separate section entitled "Related Work" to transit smoothly in the following parts of the work. Please evaluate how your study is different from others in the related work section? What you have where others do not? Why you are better or how? What's the novel here?
Reply No. 3.1:
We acknowledged the great comments in this revision and added related work at the end of the introduction. We hope that this clarifies the concerns.
Changes No. 3.1: At the end of the introduction section on page 2.
As a result, we examined 112, and Figure 1 displays the analysis plan. To date, this research has the most extensive models with seven missing patterns for quality control, four missing imputation strategies, and four power machine learning models. We aim to discover novel indications for heart failure from such a big scale of analysis models.
2)In a scientific article, it is not enough to cite the references.
The philosophy of the article is relevant to the field of ML.
The authors need to elaborate, why the traditional ML classifiers are selected. A background section about ML algorithms and techniques is necessary. Technical reasons must provide in contrast to the state-of-the-art ML approaches.
Reply No. 3.2:
We appreciate the reviewer's comment very much. We not only gave citations but also wrote brief descriptions for the methods in the previous revision.
.
In the introductions, we had the following information.
Machine learning equips with algorithms that improve performance with experience [5]. A machine learns if its performance at tasks improves with experience [6]. In constructing a model using machine learning techniques, the first step divides the entire data into two parts: the training data and the testing data. The training data refers to the dataset used to train the machine learning hyper-parameters. The optimized model parameters are estimated based on the training datasets. The testing data is independent of the training data. The testing data will not participate in the training process but only be used for evaluations of the trained model. To avoid over-fitting [7], K-Fold Cross Validation (CV) is a common strategy. The CV errors based on accuracy, mean square error (MSE), and F1 Score are adopted [8].
Although it has indicated that African Americans had a higher risk of incident heart failure among all populations [3], few studies have developed the risk prediction model for heart failure in African-Americans. Therefore, this study utilized the Jackson Heart Study (JHS), the most prominent African-American research database in the United States, and machine learning methods to construct comprehensive predictive models for heart failure. Our large-scale analysis structure aims to discover missing and yet important information in such a big-data approach.
Random Forest could provide Feature Importance through Decision Trees [9]. Breiman discussed the complete algorithm in 2001 [10], a type of ensemble method that collects multiple weak classifiers that produce a robust classifier [11]. Classification And Regression Trees (CART) is a decision tree for predictive classification and continuous value [12] that adopts the binary division rule. Each time, a division generates two branches, and the Gini classification method determines which branch is the best. Extreme Gradient Boosting (XGBoost) is a further extension from GBDT (Gradient Boosted Decision Tree) [13]
In the whole methods section, we did provide many detailed descriptions for our models.
In this second revision, we had added more details for the methods.
Changes No. 3.2: The last paragraph of the introduction section on page 2 has been modified as the following.
After missing data are correctly imputed, four methods, including (1) Lasso logistic regression, (2) Support Vector Machine (SVM) [6, 7], (3) random forest, and (4) Extreme Gradient Boosting (XGBoost), would be constructed for a predictive model for heart failure based on the African-Americans population. The inclusion of the four strategies considers the essential aspects of the analysis concepts. Lasso logistic regression is the conventional statistical approach with a penalty term that avoids overfitting in regression models. The SVM is a machine learning tool dealing with classification problems. The SVM classifies subjects according to the separating hyperplane, which is defined as , where and . The last two methods are tree-based machine learning. However, the XGBoost depends on gradient boosting trees and the objection function is defined as , where is the sum of the first derivative of the loss function in the jth leaf and represents similar calculations for the second derivative.
3)There is no clear presentation of the results and their commentary
Τhe authors continuously project figures and Tables without highlighting the methodology and the importance of the results. I recommend the authors discuss the merits of the proposed results.
Reply No 3.3:
The importance message is well taken.
Changes No. 3.3: We added the following paragraph on page 7.
According to these results, our analysis plan is an excellent example for future studies in various health outcomes. We conclude that even when variables have a missing rate as high as 30%, as long as we adopt the random forest imputation before the analysis, the XGBoost provides the best predictive ability.
Authors should provide adequate explanations for their methodology in the article.
In a scientific article, each technique should be compared with already known methods, and its superiority should be documented. That is the philosophy of the research. Also, the more details are given about the proposed approach, the better for the reader.
Reply No 3.4:
We thank the reviewer for such a great comment. We agree that we did not develop any new analysis model but provide comprehensive combinations of analysis models. In our first reply, we had emphasized our contribution, and the finding of diabetes medications as the most important feature in machine learning strategies is novel. We tried to give the most succinct message to the readers. Since the reviewer suggests detailed results, we added supplement materials in the second revision.
Changes No. 3.4: Supplement materials are included in the second revision.
